# Clinical Characteristics and Predictors of In-Hospital Mortality of Patients Hospitalized with COVID-19 Infection

**DOI:** 10.3390/jcm12010143

**Published:** 2022-12-24

**Authors:** Leszek Gromadziński, Maciej Żechowicz, Beata Moczulska, Michał Kasprzak, Klaudyna Grzelakowska, Paulina Nowek, Dominika Stępniak, Natalia Jaje-Rykowska, Aleksandra Kłosińska, Mikołaj Pożarowszczyk, Aleksandra Wochna, Adam Kern, Jerzy Romaszko, Agata Sobacka, Przemysław Podhajski, Aldona Kubica, Jacek Kryś, Maciej Piasecki, Piotr Lackowski, Małgorzata Jasiewicz, Eliano Pio Navarese, Jacek Kubica

**Affiliations:** 1Department of Cardiology and Internal Medicine, School of Medicine, Collegium Medicum, University of Warmia and Mazury in Olsztyn, 10-082 Olsztyn, Poland; 2Collegium Medicum, Nicolaus Copernicus University, 85-094 Bydgoszcz, Poland; 3School of Medicine, Collegium Medicum, University of Warmia and Mazury in Olsztyn, 10-082 Olsztyn, Poland; 4Department of Family Medicine and Infectious Diseases, School of Medicine, Collegium Medicum, University of Warmia and Mazury in Olsztyn, 10-082 Olsztyn, Poland

**Keywords:** mortality, SARS-CoV-2 infection, COVID-19, prognosis, risk factors

## Abstract

Background: The identification of parameters that would serve as predictors of prognosis in COVID-19 patients is very important. In this study, we assessed independent factors of in-hospital mortality of COVID-19 patients during the second wave of the pandemic. Material and methods: The study group consisted of patients admitted to two hospitals and diagnosed with COVID-19 between October 2020 and May 2021. Clinical and demographic features, the presence of comorbidities, laboratory parameters, and radiological findings at admission were recorded. The relationship of these parameters with in-hospital mortality was evaluated. Results: A total of 1040 COVID-19 patients (553 men and 487 women) qualified for the study. The in-hospital mortality rate was 26% across all patients. In multiple logistic regression analysis, age ≥ 70 years with OR = 7.8 (95% CI 3.17–19.32), *p* < 0.001, saturation at admission without oxygen ≤ 87% with OR = 3.6 (95% CI 1.49–8.64), *p* = 0.004, the presence of typical COVID-19-related lung abnormalities visualized in chest computed tomography ≥40% with OR = 2.5 (95% CI 1.05–6.23), *p* = 0.037, and a concomitant diagnosis of coronary artery disease with OR = 3.5 (95% CI 1.38–9.10), *p* = 0.009 were evaluated as independent risk factors for in-hospital mortality. Conclusion: The relationship between clinical and laboratory markers, as well as the advancement of lung involvement by typical COVID-19-related abnormalities in computed tomography of the chest, and mortality is very important for the prognosis of these patients and the determination of treatment strategies during the COVID-19 pandemic.

## 1. Introduction

COVID-19 is a lung disease caused by infection with the SARS-CoV-2 virus. The disease is novel, as it emerged in December 2019, and has caused over 6 million deaths worldwide thus far. The most affected regions in the world are Europe with over 194 million cases, Asia with over 149 million, and North America with over 99 million. In addition, Europe is the region with the highest death toll due to COVID-19 with over 1.8 million deaths. According to the available literature, elderly people with chronic cardiovascular diseases and diabetes are at a high risk of severe COVID-19 [1,2]. Despite differences between countries, the overall in-hospital mortality due to COVID-19 is high, amounting to 30% [3]. Thus far, the clinical picture of the disease indicating the worst prognosis has not been clearly determined. The reasons underlying the different clinical courses of the disease among patients are still unknown; therefore, it is difficult to predict which patients will develop a severe condition and which will present with only a mild form of the disease.

Hence, the identification of mortality predictors is crucial not only for adjusting the clinical approach but also for the monitoring and intensification of the therapy. Among the clinical parameters, the proposed predictors of poor prognosis include age, comorbidities, and male sex [4]. Moreover, a number of laboratory parameters have been associated with mortality in various studies. The highest risk of death has been observed for routine blood tests (lymphocytes, leukocytes, neutrophils, platelets, and hemoglobin), coagulation indices (D-dimer and prothrombin time), markers of liver and kidney function, inflammatory factors, and cardiac troponins [5,6,7,8,9,10]. However, medical data concerning the factors influencing the course, severity, and consequences of COVID-19 have still not been fully examined.

Validation of these early reports could equip clinicians with a valuable tool for clinical risk stratification. Therefore, in our study, we assessed independent factors of in-hospital mortality in a group of 1040 COVID-19 patients from two medical centers, admitted during the second wave of the pandemic.

## 2. Material and Methods

### 2.1. Study Design and Participants

The data were obtained from a prospective registry of patients with COVID-19. We included 1040 patients admitted to the COVID-19 unit of the University Hospital in Olsztyn (Hospital A) and the COVID-19 unit of the University Hospital in Bydgoszcz (Hospital B), between October 2020 and May 2021. All patients were diagnosed with SARS-CoV-2 infection via a positive RT-PCR assay of nasopharyngeal swabs. The present study was approved as a prospective observational protocol by the local ethics committee (protocol no. 24/2021).

In this study, we compared various baseline demographic and clinical characteristics along with laboratory data and clinical outcomes in both survivor and non-survivor COVID-19-confirmed patients admitted to two medical centers. The study flowchart is illustrated in Figure 1.

### 2.2. Data Collection

The demographic, clinical, and laboratory data were retrieved from individual clinical records and recorded in a dedicated electronic database. For all patients included in the study, we assessed the following comorbidities: arterial hypertension (AH), coronary artery disease (CAD), chronic heart failure (CHF), diabetes mellitus (DM), chronic obstructive pulmonary disease (COPD), malignancies, obesity, and clinical data: heart rate at admission (HR), systolic blood pressure (SBP), diastolic blood pressure (DBP), duration of hospitalization, and smoking status. In the analysis, apart from sex and age, we included the following laboratory parameters at hospital admission: C-reactive protein (CRP), procalcitonin, alanine aminotransferase (ALT), aspartate aminotransferase (AST), coagulation (D-dimer, APTT, PT, INR), complete blood count with differential (WBC—white blood cells, RBC—red blood cells, MCV—mean corpuscular volume, Hb—hemoglobin, Ht—hematocrit, PLT—platelets, lymphocytes, RDW—red cell distribution width, PDW—platelets distribution width, MPV—mean platelet volume), glucose, creatinine, eGFR, urea, potassium, sodium, ferritin, cardiovascular biomarkers—namely N-terminal pro-B-type natriuretic peptide (NT-proBNP) and troponin T (Tn-T), saturation at admission without oxygen (SpO_2_), oxygenation index (partial pressure of oxygen to inspiratory fraction of oxygen ratio (PaO_2_/FiO_2_)), pO_2_—partial pressure of oxygen, pCO_2_—partial pressure of carbon dioxide, and blood pH level.

Moreover, all patients underwent computed tomography (CT) of the chest with a lung involvement assessment defined as a percentage of the lungs affected by typical COVID-19-related abnormalities.

Specific in-hospital treatments consisted of antiviral therapy with remdesivir, enoxaparin, steroids, and tocilizumab. If needed, patients were treated with antibiotics and oxygen support in the form of low-flow (nasal cannula) and high-flow (Venturi and reservoir masks, nasal high flow) oxygen, as well as noninvasive (NIV) and invasive ventilation by a ventilator.

### 2.3. Statistical Analyses

For comparison, a Student’s *t* test or Mann–Whitney test was used depending on the distribution of variables. Demographic and clinical data were expressed as the mean and standard deviation when normally distributed, otherwise by median and first-third quartiles. Categorical variables, whenever dichotomous or nominal, were reported as frequencies and percentages and analyzed with the application of the Chi-square test. 

Univariate and multivariate Cox proportional hazard models were used to evaluate the estimate of the hazard ratios (HR) and 95% confidence intervals (CI) for the correlation between all variables and in-hospital mortality. For multivariate analysis, only those parameters that demonstrated statistical significance (*p* < 0.1) in univariate analysis were included. 

A receiver operating characteristic (ROC) curve was drawn for the laboratory values that were statistically correlated with mortality, and the value with the highest sum of sensitivity and specificity was accepted as the cut-off value. 

## 3. Results

### 3.1. Population Characteristics

A total of 1040 COVID-19 patients (553 men and 487 women) qualified for the study. The median age was 70 (61–80) years. The majority of the patients, namely 770 (74.0%), were discharged alive, while the remaining 270 (26.0%) died. Among all patients included in the study, only 63 individuals were vaccinated (6.0%), and only 18 were vaccinated with the full dose (1.7%). The mortality rate in all vaccinated patients was two times lower than in unvaccinated patients and amounted to 8 deaths out of 63 patients (12.7%).

### 3.2. Biochemical Biomarkers and Clinical Features as Independent Predictors of In-Hospital Mortality

The demographic and clinical characteristics of patients in both groups are presented in Table 1. 

Non-survivors were significantly older, more likely to have AH, CAD, CHF, and DM, and had a lower SBP and DBP, slower HR, and a shorter hospital stay than patients who survived. The analyzed laboratory findings are presented in Table 2.

Non-survivor patients were characterized by higher levels of AST, CRP, D-dimer, PT, potassium, glucose, creatinine, urea, WBC, neutrophils, RDW-CV, NT-proBNP, INR, procalcitonin, troponin-T, and ferritin, and by lower levels of pH, eGFR, RWC, PLT, lymphocytes, SpO_2_ at admission, and oxygenation index (PaO_2_/FiO_2_) as compared with survivor patients. By contrast, there were no significant differences between survivors and non-survivors as concerns sex, BMI, smoking status, COPD, malignancies, ALT, APTT, pCO_2_, pO_2_, hemoglobin, hematocrit, PDW, MPV, and sodium.

### 3.3. Medications Associated with In-Hospital Mortality

A significant difference between survivors and non-survivors was observed regarding treatment, as presented in Table 3. 

Patients who died received oxygen (*p* < 0.0001), antibiotics (*p* < 0.0001), and steroids (*p* < 0.0001) more often, while receiving convalescent plasma (*p* < 0.0001) and remdesivir (*p* < 0.0001) less often. Moreover, in the chest CT, the lung involvement expressed as a percentage of the lungs affected by typical COVID-19-related abnormalities was significantly higher (*p* < 0.0001) in non-survivor patients as compared to survivors.

In order to assess independent risk factors for in-hospital mortality of COVID-19 patients, univariate and multivariate analyses of Cox proportional hazards regression were performed. The potential significance of clinical and laboratory parameters was analyzed. 

### 3.4. Risk Factors for Death in COVID-19 Patients

The potential risk factors for death in COVID-19 patients are presented in Table 4. 

However, in a multivariate analysis, the only independent risk factors for death were age ≥ 70 years with OR = 7.8 (95% CI 3.17–19.32), *p* < 0.001, presence of CAD with OR = 3.5 (95% CI 1.38–9.10), *p* = 0.009, SpO_2_ ≤ 87% with OR = 3.6 (95% CI 1.49–8.64), *p* = 0.004, and typical COVID-19-related lung abnormalities in the chest CT ≥ 40% with OR = 2.5 (95% CI 1.05–6.23), *p* = 0.037 (Table 5).

A ROC was created for demographic and laboratory parameters. It was determined that age ≥ 70 years, SpO_2_ ≤ 87%, and typical COVID-19-related lung abnormalities in the chest CT ≥ 40% predicted mortality. The results for different cut-off values of these parameters are provided in Table 6.

## 4. Discussion

The COVID-19 pandemic has spread fear not only because of its enormous death toll worldwide, but also due to its unpredictable course and unknown risk factors for severity, need for hospitalization, and mortality. There has been a growing number of publications concerning different parameters, which could potentially influence further diagnostic and treatment guidelines. However, the low sample sizes and heterogeneity of previous studies have strongly impacted the consolidation of the primary results [11,12]. Based on a large group of patients, we highlight clinical, biochemical, and radiographic features and markers related to severe outcomes or death due to COVID-19 [13,14]. 

Many of the early publications originate from Asia and North America and were intended to create a useful tool for risk stratification based on some multicenter assessments during the first pandemic wave [15]. The advantage of our observation is the positioning of the study in the second wave of the pandemic in Poland (between October 2020 and May 2021), which enabled us to follow more consistent guidelines. Hence, our study population appears to be more homogenous and representative regarding high and very high severity and death risk due to COVID-19 [16,17]. Our prospective registry of the study population probed deeper by including relevant symptoms, vital signs, and easy-to-obtain respiratory parameters such as SpO_2_ (oxygen saturation at admission) and PaO_2_/FiO_2_ (oxygenation index) [18]. Such multifactor analysis and stratification appear to be consistent with the global systematic review and meta-analysis by Booth et al. published in 2021, which included over 150,000 patients. The methodology and choice of publications for this meta-analysis demonstrated a very high level of inconsistency between the previous studies (less than 1% of primarily considered patients were analyzed) with a geographical dominance of China [19]. Only one publication from Poland was included in this meta-analysis; however, it referred to the first wave of the pandemic [20]. The latest data from Polish centers were unavailable at that time.

All patients included in our study were diagnosed via a positive RT-PCR assay of nasopharyngeal swabs and underwent CT of the chest at admission to evaluate the extent of the inflammation typical of COVID-19 pneumonia. This approach was considered most valuable as regards sensitivity and specificity [21]. Unfortunately, such a combination was not widely available in many countries, where one could find the chest X-ray (CXR) with its visual CXR score to be useful [22]. In our study group, the CT scan result with the cut-off level of 40% lung involvement appears to be a strong univariate predictor of in-hospital mortality (OR 3.95, 95% CI 2.34–4.65, *p* < 0.0001), as well as an independent risk factor in multivariate analysis (OR 2.5, 95% CI 1.05–6.23, *p* = 0.037). In a number of smaller-size studies, the chest CT score in COVID-19 patients correlated highly with biochemical findings and disease severity while being useful for short-term prognosis [23,24]. Furthermore, according to Colombi et al., qualitative and quantitative chest CT parameters obtained visually or by software are predictors of mortality and proven to be better predictors of death compared with clinical models [25]. 

Our results seem to undermine the advantage of radiography over a clinical model. First, our study population was about five-fold larger than the one cited above. Second, other independent risk factors for death due to COVID-19 pneumonia in multivariate analysis were demographic and clinical: age over 70 years (OR 7.8, 95% CI 3.17–19.32, *p* < 0.001), presence of CAD (OR 3.5, 95% CI 1.38–9.10, *p* = 0.009), or SpO_2_ ≤ 87% (OR 3.6, 95% CI 1.49–8.64, *p* = 0.004). Additionally, when analyzing the results of the univariate predictors, both biochemical and clinical factors were significant (Table 4). These findings align with the previous smaller-size studies and systematic reviews, highlighting that age and multiple comorbidities (hypertension, CAD, heart failure, diabetes) increase the risk of adverse outcomes. There were also various laboratory test abnormalities considered significant (increased levels of CRP, AST, creatinine, glucose, NT-proBNP, TnT) [26,27]. 

Special attention should be given to “respiratory” univariate predictors of in-hospital mortality: SpO_2_ at admission (*p* < 0.0001) and PaO_2_/FiO_2_ (*p* < 0.0001), which are easy to obtain and were significantly lower in the group of non-survivors. As expected, the need for more intensive and prolonged oxygen therapy was significantly higher in the group with poor outcomes (*p* < 0.0001). Grasselli et al. presented similar findings: low PaO_2_/FiO_2_ was a strong independent risk factor (OR 0.80, 95% CI 0.74–0.87) associated with a higher mortality rate in a group of almost 4000 patients hospitalized in intensive care units due to COVID-19 pneumonia. Interestingly, a history of chronic obstructive pulmonary disease (COPD) in the aforementioned group of patients (OR 1.68, 95% CI 1.28–2.19), as well as in the previously mentioned meta-analysis by Booth et al., contributed to a poor outcome, whereas in our population there were no statistically significant differences between survivors and non-survivors regarding COPD (*p* = 0.67) [28]. 

Most data are conflicting regarding the impact of tobacco smoking in patients with pneumonia due to SARS-CoV-2 infection. There are numerous meta-analyses highlighting that especially present smokers are more vulnerable to severe COVID-19 and worse in-hospital outcomes [29,30]. Conversely, there are retrospective large cohort studies indicating that neither current nor former smoking were associated with an increased risk of hospitalization, in-hospital mortality, ICU admission, or intubation [31]. Moreover, based on an analysis of thirteen studies from China and considering the well-established immunomodulatory effects of nicotine, Farsalinos et al. suggest that pharmaceutical nicotine should even be considered as a potential treatment option for COVID-19 [32]. In our study population, we observed no increased mortality risk in either former or current smokers, although we lacked some clear information about the real smoking status of each patient included in the study.

Treatment with remdesivir and treatment with convalescent plasma were found to be univariate predictors of lower in-hospital mortality (both *p* < 0.0001). In our opinion, this association resulted from the delay of antiviral treatment initiation—some patients were symptomatically treated at home for a longer period of time before hospital admission, while the remdesivir treatment was only available in the first 5–6 days from the onset of disease symptoms according to our local guidelines. At the same time, we found no advantage in treatment with tocilizumab (*p* = 0.4378), although there were no time restrictions regarding the appearance and aggravation of the symptoms.

This study has several limitations. First, we were not able to determine the genotypes of the SARS-CoV-2 virus. We only know that during the second wave of the pandemic, two variants of the virus dominated in Poland, the original variant, i.e., Wuhan, and the British variant, i.e., the alpha variant. Second, not all laboratory parameters were collected; in particular, for less than half of the patients, laboratory data for some biochemical parameters such as troponin-T and serum ferritin levels were not recorded, which may have led to an underestimation of their potential predictive value. Third, the sample size was limited, since it only included hospitalized patients. The design of the study did not allow us to accurately retrieve data to stage the underlying diseases, potentially lowering the net effect of each comorbidity. Furthermore, as the criteria for hospitalization due to COVID-19 are different across different institutions, an inclusion bias cannot be excluded in this regard. Finally, as this is an observational study, residual confounding factors may exist.

## 5. Conclusions

The relationship between clinical and laboratory markers, as well as the advancement of lung involvement by typical COVID-19-related abnormalities in CT of the chest, and mortality is very important for the prognosis of these patients and the determination of treatment strategies during the COVID-19 pandemic. 

In our study, the COVID-19 outbreak determined a high in-hospital mortality rate, the main clinical predictors of which were:age over 70 years old;decreased saturation at admission without oxygen below 87%;the advancement of lung involvement by typical COVID-19-related abnormalities in CT of the chest above 40%;and a concomitant diagnosis of CAD.

## Figures and Tables

**Figure 1 jcm-12-00143-f001:**
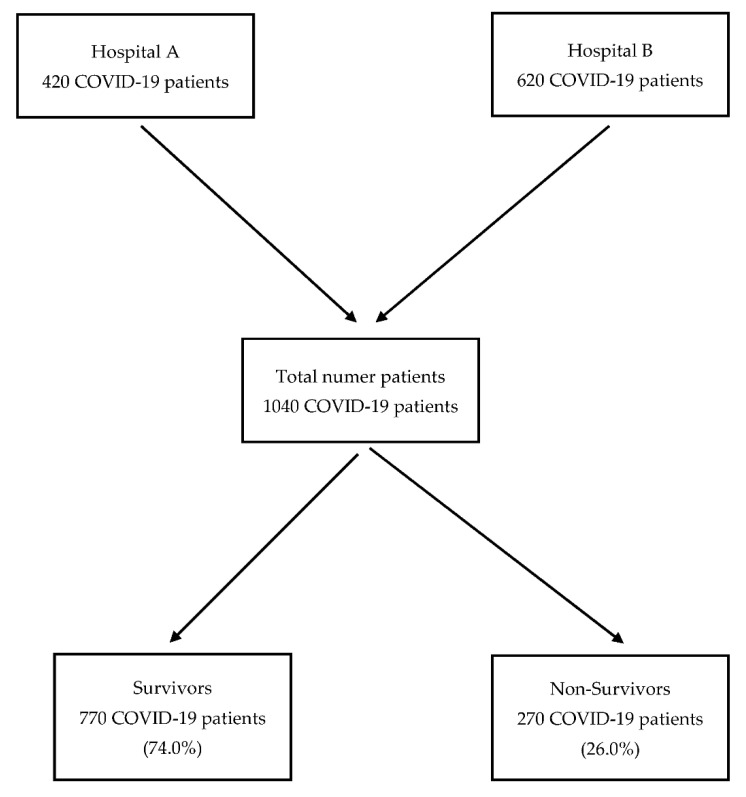
Study flowchart.

**Table 1 jcm-12-00143-t001:** General characteristics of both groups.

Parameter (n (%) or Mean ± SD)		Total (n = 1040)	Survivors (n = 770)	Non-Survivors (n = 270)
Age (years)		68.8 ± 15.4	65.7 ± 15.5	77.5 ± 11.1
Sex (n, %)	Female	487 (46.8)	362 (47)	125 (46.3)
	Male	553 (53.2)	408 (53)	145 (53.7)
Days of hospitalization (n)		13.9 ± 9.4	14.5 ± 8.5	12.3 ± 11.2
Hypertension (n, %)		643 (61.8)	454 (59)	189 (70)
CAD (n, %)		182 (17.5)	124 (16.1)	58 (21.5)
HF (n, %)		168 (16.1)	91 (11.8)	77 (28.5)
Diabetes (n, %)		329 (31.6)	222 (28.8)	107 (39.6)
Asthma, COPD (n, %)		116 (11.1)	84 (10.9)	32 (11.8)
Cancer (n, %)		48 (4.6)	30 (4.0)	18 (6.7)
Smoking (n, %)		50	44	6
Heart rate (n/min)		85.5 ± 16.5	86 ± 15.8	84.1 ± 18.2
SBP (mmHg)		130.1 ± 20.9	131.5 ± 19.6	126.1 ± 23.8
DBP (mmHg)		76.8 ± 12.8	78.1 ± 12.2	73 ± 13.6

CAD—coronary artery disease; HF—heart failure; COPD—chronic obstructive pulmonary disease; SBP—systolic blood pressure; DBP—diastolic blood pressure.

**Table 2 jcm-12-00143-t002:** Biochemical characteristics of both groups.

Parameter {Mean ± SD or Median (IQR)}, (n)	Total (n = 1040)	Survivors (n = 770)	Non-Survivors (n = 270)	*p*-Value
CRP (mg/L) (1036)	61.9 (24.2–126.7)	54.4 (19.4–119.3)	84.2 (47–164.5)	<0.0001
D-dimer (µg/L) (1001)	1030 (580–2120)	907 (510–1820)	1480.5 (860–2906)	<0.0001
ALT (U/L) (1003)	31 (19–51)	30 (20–50)	31 (18–53)	0.9706
AST (U/L) (1003)	39 (27–63)	36 (26.5–56.5)	51 (31–90)	<0.0001
APTT(s) (997)	30.9 ± 11.9	31 ± 12.4	30.8 ± 10	0.2138
pH (650)	7.43 ± 0.08	7.44 ± 0.07	7.40 ± 0.09	<0.0001
pCO_2_ (mmHg) (631)	34.95 ± 7.56	34.53 ± 6.01	35.75 ± 9.85	0.6865
pO_2_ (mmHg) (634)	68.73 ± 30.44	67.64 ± 25.32	70.79 ± 38.35	0.1451
Glucose (mg/dL) (978)	121 (102–153)	115 (99–139)	141 (114–189)	<0.0001
Creatinine (mg/dL) (1032)	0.92 (0.72–1.23)	0.86 (0.69–1.09)	1.17 (0.83–1.7)	<0.0001
eGFR (mL/min/1.73 m^2^) (1032)	72.81 ± 28.53	78.22 ± 26.86	57.15 ± 27.46	<0.0001
Urea (mg/dL) (942)	26.45 (16.6–44)	22.2 (15.2–35.4)	45.05 (29.3–75)	<0.0001
RBC (million/mm^3^) (1036)	4.29 ± 0.72	4.32 ± 0.69	4.19 ± 0.77	0.0036
WBC (thousand/µL) (1036)	6.72 (4.76–9.1)	6.3 (4.61–8.51)	7.97 (5.48–11.7)	<0.0001
Hemoglobin (g/dL) (1036)	12.93 ± 2.17	12.99 ± 2.07	12.74 ± 2.44	0.1973
Hematocrit (%) (1036)	38.02 ± 6.21	38.11 ± 5.96	37,76 ± 6,89	0.4113
MCV (fL) (1036)	88.92 ± 6.75	88.51 ± 6.59	90.10 ± 7.05	<0.0001
PLTx10^9^ per L (1036)	229.2 ± 105	235.3 ± 108.2	211.6 ± 93	0.0007
Neutrophils (%) (1021)	73.42 ± 15.1	71.1 ± 14.84	80.18 ± 13.80	<0.0001
Lymphocytes ×10^9^ per L (822)	12.95 (1.8–23.7)	15.2 (4.8–25.2)	5.3 (1–15.5)	<0.0001
RDW-CV% (1034)	14.17 ± 3.84	13.98 ± 4.24	14.75 ± 2.2	<0.0001
PDW% (1027)	13.1 ± 3.53	13.01 ± 3.74	13.37 ± 2.80	0.0625
MPV% (1027)	10.91 ± 3.07	10.91 ± 3.50	10.9 ± 1.16	0.1973
NT-proBNP (pg/mL) (855)	632 (186.9–2372)	438.9 (144.5–1596)	2253.5 (641.1–7062)	<0.0001
INR (1013)	1.15 (1.06–1.28)	1.15 (1.06–1.26)	1.2 (1.07–1.34)	0.0029
PT(s) (1010)	12.8 (11.5–14.2)	12.7 (11.5–14)	12.9 (11.5–14.7)	0.0273
Potassium (mg/dL) (1033)	4.06 ± 0.66	4 ± 0.61	4.22 ± 0.76	<0.0001
Procalcitonin (µg/L) (891)	0.11 (0.06–0.29)	0.09 (0.05–0.2)	0.29 (0.12–0.79)	<0.0001
Sodium (mg/dL) (1033)	137.97 ± 5.64	137.69 ± 5.12	138.76 ± 6.88	0.1502
TSH (mlU/L) (548)	0.96 (0.52–1.62)	0.98 (0.58–1.63)	0.83 (0.36–1.59)	0.0735
Troponin T (ng/L) (315)	20 (11–43)	14 (9–27)	38.5 (23–79)	<0.0001
Ferritin (µg/L) (535)	411 (204–819)	379.5 (192.5–755.5)	551 (275–1500)	0.0042
Computed tomography (%) (839)	25 (10–45)	20 (10–40)	40 (15–70)	<0.0001
Serum ferritin (µg/L) (535)	411 (204–819)	379.5 (192.5–755.5)	551 (275–1500)	0.0042
SpO_2_ (%) (826)	90.31 ± 8.44	91.63 ± 6.55	85.04 ± 12.29	<0.0001
PaO_2_/FiO_2_ (mmHg) (1026)	276.2 (166.7–380.9)	304.8 (216.7–428.6)	186.7 (76.8–276.2)	<0.0001

RBC—red blood cells; WBC—white blood cells; CRP—C-reactive protein; PCT—procalcitonin; AST—aspartate transaminase; ALT—alanine transaminase; APTT –- activated partial thromboplastin time; PT—prothrombin time; NT-proBNP—N-terminal prohormone of brain natriuretic peptide; RDW—red blood cell distribution width; PDW—platelet distribution width; MPV—mean platelet volume; SpO_2_—oxygen saturation at admission, (PaO_2_/FiO_2_)—oxygenation index.

**Table 3 jcm-12-00143-t003:** Treatment in both groups.

Parameter {n (%)}	Total (n = 1040)	Survivors (n = 770)	Non-Survivors (n = 270)	*p*-Value
Remdesivir (n, %)	178 (17.1)	154 (20)	24 (8.9)	<0.0001
Tocilizumab (n, %)	92 (8.8)	65 (8.4)	27 (10)	0.4378
Antibiotic treatment (n, %)	952 (91.5)	684 (88.8)	268 (99.2)	<0.0001
Anticoagulants (n, %)	1006 (96.7)	743 (96.5)	263 (97.4)	0.4675
Steroids (n, %)	845 (81.3)	599 (77.8)	246 (91.1)	<0.0001
Convalescent plasma (n, %)	315 (30.3)	263 (34.2)	52 (19.3)	<0.0001
Oxygen therapy (n, %)	929 (89.3)	661 (85.8)	268 (99.2)	<0.0001
NIV (n, %)	131 (12.6)	64 (8.3)	67 (24.8)	<0.0001
Ventilator therapy (n, %)	160 (15.4)	33 (4.3)	127 (47)	<0.0001

NIV—noninvasive ventilation.

**Table 4 jcm-12-00143-t004:** Univariate predictors of in-hospital mortality of patients with COVID-19.

Parameter	OR	−95% CI	+95% CI	*p*-Value
Age (for every year)	1.066	1.053	1.08	<0.0001
Age ≥ 70	4.408	3.232	6.011	<0.0001
Days of hosp. (n)	0.968	0.95	0.987	0.0008
AST (U/L)	1.006	1.003	1.009	<0.0001
CRP (mg/L)	1.005	1.003	1.007	<0.0001
D-dimer (µg/L)	1.025	1.008	1.042	0.0035
pH	0.0026	0.0002	0.029	<0.0001
pCO_2_ (mmHg)	1.021	0.999	1.043	0.0599
Glucose (mg/dL)	1.007	1.005	1.009	<0.0001
Creatinine (mg/dL)	1.315	1.157	1.493	<0.0001
eGFR (mL/min/1.73 m^2^)	0.974	0.969	0.979	<0.0001
Urea (mg/dL)	1.027	1.022	1.033	<0.0001
RBC (million/mm^3^)	0.769	0.634	0.933	0.0078
WBC (thousand/µL)	1.086	1.056	1.117	<0.0001
MCV (fL)	1.039	1.016	1.063	0.0009
PLT × 10^9^ per L	0.998	0.997	0.999	0.0065
Neutrophils (%)	1.054	1.04	1.067	<0.0001
Lymphocytes × 10^9^ per L	0.975	0.962	0.989	0.0003
RDW-CV%	1.078	1.016	1.145	0.0134
NT-proBNP (pg/mL)	1.097	1.067	1.128	<0.0001
Potassium (mg/dL)	1.64	1.328	2.025	<0.0001
Procalcitonin (µg/L)	1.017	0.997	1.038	0.0943
Sodium (mg/dL)	1.035	1.009	1.061	0.0078
Computed tomography (%)	1.027	1.02	1.034	<0.0001
Computed tomography ≥ 40%	3.295	2.336	4.649	<0.0001
Serum ferritin (µg/L)	1.00036	1.00015	1.00058	0.001
SBP (mmHg)	0.988	0.982	0.995	0.0006
DBP (mmHg)	0.969	0.958	0.98	<0.0001
SpO_2_ (%)	0.923	0.904	0.942	<0.0001
SpO_2_ ≤ 87%	4.625	3.152	6.786	<0.0001
PaO_2_/FiO_2_ (mmHg)	0.994	0.993	0.995	<0.0001
Hypertension	1.624	1.206	2.186	0.0014
CAD	1.538	1.091	2.17	0.0141
HF	3.198	2.281	4.483	<0.0001
Type 2 diabetes	1.735	1.31	2.298	0.0001
Oxygen therapy	18.45	5.834	58.341	<0.0001
Antibiotic treatment	7.185	3.853	13.397	<0.0001
Steroids	2.556	1.629	4.01	<0.0001
Convalescent plasma	0.44	0.319	0.606	<0.0001
Remdesivir	0.341	0.223	0.522	<0.0001
NIV	4.089	2.812	5.945	<0.0001
Ventilator therapy	21.749	14.29	33.101	<0.0001

RBC—red blood cells; WBC—white blood cells; CRP—C-reactive protein; PCT—procalcitonin; AST—aspartate transaminase; ALT—alanine transaminase; APTT—-activated partial thromboplastin time; PT—prothrombin time; NT-proBNP—N-terminal prohormone of brain natriuretic peptide; RDW—red blood cell distribution width; CAD—coronary artery disease; HF—heart failure; SBP—systolic blood pressure; DBP—diastolic blood pressure; SpO_2_—oxygen saturation at admission; PaO_2_/FiO_2_—oxygenation index; NIV—noninvasive ventilation.

**Table 5 jcm-12-00143-t005:** Multivariate predictors of in-hospital mortality of patients with COVID-19.

Parameter	OR	−95% CL	+95% CL	*p*-Value
Age ≥ 70	7.824	3.168	19.321	<0.001
CAD	3.544	1.38	9.105	0.009
SpO_2_ ≤ 87%	3.589	1.49	8.646	0.004
Computed tomography ≥ 40%	2.567	1.056	6.236	0.037

CAD—coronary artery disease, SpO_2_—oxygen saturation at admission.

**Table 6 jcm-12-00143-t006:** Receiver operating characteristics analysis of demographic and laboratory parameters of in-hospital mortality of patients with COVID-19.

Parameter	AUC	95% CI	*p*
Age ≥ 70 years	0.729	0.696–0.762	*p* < 0.0001
SpO_2_ ≤ 87%	0.69	0.642–0.738	*p* < 0.0001
Computed tomography ≥ 40%	0.672	0.626–0.718	*p* < 0.0001

SpO_2_—oxygen saturation at admission.

## Data Availability

The data presented in this study are available on request from the corresponding author.

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
