# Peer review of "Clinical Characteristics and Predictors of In-Hospital Mortality of Patients Hospitalized with COVID-19 Infection"

_jcm, 2022, doi:10.3390/jcm12010143_

Round 1
Reviewer 1 Report
Gromadzinski L. et al reported the characteristics and predictors of in-hospital mortality of patients hospitalized with COVID-19 infection. They included patients from two hospitals with the diagnosis of COVID-19 between October 2020 and May 2021 and analyzed the clinical and demographic features, the presence of comorbidities, laboratory parameters, and radiological findings at admission with relationship with mortality. They found that the clinical and laboratory markers as well as advancement of lung involvement by typical COVID-19 changes in CT of the chest are related to mortality, which is very important to prognose these patients. The results are interesting and have important role in application for clinical diagnose and treatment of COVID-19. However, some issues are still needed to addressed:
1. The patients included are older than 50. It is better to expand the age range, which may help for the conclusion of age >70 may associate with mortality of COVID-19.
2. As we all know, different genotypes of SARS-CoV-2 has different ability of infection and pathogenesis. The authors should also analyze the genotypes influence on the mortality.
3. It is better to follow up these markers and analyze their relationship with patients’ pathogenesis.
4. Please check and modify the format of the references one by one. For example: ref.1, Nature Microbiology should be used with its abbreviated format like “Nat Microbiol”.
Author Response
Authors would like to thank the Reviewer for the opinion regarding our work. We are very grateful for all the suggestions and comments that were more than valuable and allowed us to improve the manuscript. Please find below our comments according to your remarks.
According to the reviewer's suggestions the manuscript has been undergone extensive English revisions.
- The patients included are older than 50. It is better to expand the age range, which may help for the conclusion of age >70 may associate with mortality of COVID-19.
Response 1. Of course, we included patients of all ages from 18 to 100 years old. In our group, 159 patients were under the age of 50, which is over 15% of the entire population.
2. As we all know, different genotypes of SARS-CoV-2 has different ability of infection and pathogenesis. The authors should also analyze the genotypes influence on the mortality.
Response 2. We agree with the reviewer's remark, but unfortunately this is a limitation of our article, as we were not able to determine the genotypes of the SARS-CoV-2 virus. We only know that during the second wave of the pandemic, two variants of the virus dominated in Poland, the original variant, i.e. Wuhan, and the British variant, i.e. the alpha variant. We will write this point in the limitation’s study.
3. It is better to follow up these markers and analyze their relationship with patients’ pathogenesis.
Response 3. Of course, we agree with the reviewer's suggestion, we hope that the discussion has been properly improved.
4. Please check and modify the format of the references one by one. For example: ref.1, Nature Microbiology should be used with its abbreviated format like “Nat Microbiol”.
Response 4. Thank you very much for your attention, the references have been modified in accordance with the reviewer's recommendations.
We hope that after changes as recommended by the reviewer, our manuscript would now be suitable for publication in the Journal of Clinical Medicine.
Reviewer 2 Report
The authors tried to study clinical characteristics and predictors of in-hospital mortality of patients hospitalized with COVID-19 infection in their study. It is very good article while methodology results sections are difficult to read. It would be better to add subheadings in both sections that would be easy for the readers. Also, I recommend the authors to add a graph within the methodology sections to illustrate the study plan and stages. Discussion is too long and there are too much redundancy. The authors need to add the limitations in this study.
Author Response
Authors would like to thank the Reviewer for the opinion regarding our work. We are very grateful for all the suggestions and comments that were more than valuable and allowed us to improve the manuscript. Please find below our comments according to your remarks.
According to the reviewer's suggestions the manuscript has been undergone extensive English revisions.
The authors tried to study clinical characteristics and predictors of in-hospital mortality of patients hospitalized with COVID-19 infection in their study. It is very good article while methodology results sections are difficult to read. It would be better to add subheadings in both sections that would be easy for the readers. Also, I recommend the authors to add a graph within the methodology sections to illustrate the study plan and stages. Discussion is too long and there are too much redundancy. The authors need to add the limitations in this study.
Response. Thank you for the thorough review and significant remarks. We reorganized the methodology results sections as indicated. The discussion has been reviewed and its excess has been reduced accordingly without changing its main stream and not reducing the main content or limiting the relevant publications. The study flowchart has been attached to the methodology sections
We hope that after changes as recommended by the reviewer, our manuscript would now be suitable for publication in the Journal of Clinical Medicine.
Reviewer 3 Report
The current manuscript is a very valuable study because it deals with the issues that can potentially save many lives of current and future corona-patients. i have only small comments. 1. i would very much like to see for this data-set also the effects of previous corona infections and of the number of vaccinations of patients. do previous corona infections measurably increase the probability of a lethal outcome or does a larger number of vaccine shots taken by people decrease the lethality measurably. such statements---if statistically true---would be extremely valuable. if this is not the case, this fact should also be clearly reported in the revised version. 2. the main conclusions of the manuscript are to be summarized in the end, as an itemized list of clear statements which are to be understandable also without reading the entire text.
Author Response
Authors would like to thank the Reviewer for the opinion regarding our work. We are very grateful for all the suggestions and comments that were more than valuable and allowed us to improve the manuscript. Please find below our comments according to your remarks.
According to the reviewer's suggestions the manuscript has been undergone extensive English revisions.
The current manuscript is a very valuable study because it deals with the issues that can potentially save many lives of current and future corona-patients. i have only small comments. 1. i would very much like to see for this data-set also the effects of previous corona infections and of the number of vaccinations of patients. do previous corona infections measurably increase the probability of a lethal outcome or does a larger number of vaccine shots taken by people decrease the lethality measurably. such statements if statistically true would be extremely valuable. if this is not the case, this fact should also be clearly reported in the revised version. 2. the main conclusions of the manuscript are to be summarized in the end, as an itemized list of clear statements which are to be understandable also without reading the entire text.
Response. Thank you very much for your attention, the sentences below will be included in the results section, the conclusion section has also been changed as suggested by the the reviewer's recommendations.
„Among all patients included in the study, only 63 individuals were vaccinated (6.0%), and only 18 were vaccinated with the full dose (1.7%). The mortality rate in all vaccinated patients was twice lower than in unvaccinated patients and amounted to 8 deaths out of 63 patients (12.7%)”
Due to the small percentage of vaccinated patients and the majority of them vaccinated with the incomplete dose, this group of patients was not analyzed in comparison to the unvaccinated group.
But these preliminary results already support the importance of vaccination in reducing mortality in SARS-CoV-2 patients.
We hope that after changes as recommended by the reviewer, our manuscript would now be suitable for publication in the Journal of Clinical Medicine.
Round 2
Reviewer 1 Report
I don't have any comments on it.